

# A diatom extension to the cGEnIE Earth system model – EcoGEnIE 1.1

Aaron A. Naidoo-Bagwell[1], Fanny M. Monteiro[2], Katharine R. Hendry[3,4], Scott Burgan[5], Jamie D. Wilson[3,6], Ben A. Ward[7], Andy Ridgwell[8] and Daniel J. Conley[1]

[1]Department of Geology, Lund University, Lund, Sweden
[2]School of Geographical Sciences, BRIDGE, University of Bristol, Bristol, UK
[3]School of Earth Sciences, University of Bristol, Bristol, UK
[4]British Antarctic Survey, Cambridge, UK
[5]Met Office, Exeter, UK
[6]Department of Earth, Ocean and Ecological Sciences, University of Liverpool, UK
[7]School of Ocean and Earth Science, University of Southampton, Waterfront Campus, Southampton, UK
[8]Department of Earth and Planetary Sciences, University of California, Riverside, California, USA

*Correspondence to*: Aaron Naidoo-Bagwell (aaron.naidoo-bagwell@geol.lu.se) and Fanny Monteiro (f.monteiro@bristol.ac.uk)

**Abstract.** We extend the ecological component ('ECOGEM') of the carbon-centric Grid Enabled Integrated Earth system model ('cGEnIE') to include a diatom functional group. ECOGEM represents plankton community dynamics via a spectrum of ecophysiological traits originally based on size and plankton food web (phyto- and zooplankton; EcoGEnIE 1.0), which we developed here to account for a diatom functional group (EcoGEnIE 1.1). We tuned EcoGEnIE 1.1, exploring a range of ecophysiological parameter values specific to phytoplankton, including diatom growth and survival (18 parameters over 250 runs) to fit best the model behaviour akin to observations of diatom biogeography, size class distribution, and global ocean biogeochemistry. This, in conjunction with a previously developed representation in the water column of opal dissolution and an updated iron cycle, produced an improved distribution of dissolved oxygen in the water column relative to the previous EcoGEnIE 1.0 as well as a value for global export production (7.5 Pg C yr$^{-1}$) closer to previous estimates. Simulated diatom biogeography is characterised by larger size classes dominating at high latitudes, notably in the Southern Ocean, and smaller size classes dominating at lower latitudes. Overall, diatom biological productivity accounts for ~ 20% of global carbon biomass in the model, with diatoms out-competing other phytoplankton functional groups when dissolved silica is available due to their faster maximum photosynthetic rates and reduced palatability to grazers. Adding a diatom functional group provides the cGEnIE Earth system model with an extended capability to explore ecological dynamics and their influence on ocean biogeochemistry through the late Mesozoic and Cenozoic, as well as enabling a broader range of palaeoceanographic proxies to be interpreted.

## 1 Introduction

Dissolved silica (dSi) – $H_4SiO_4$ (orthosilicic acid) – plays a key role in numerous biogeochemical cycles, particularly in marine environments. Marine silicifiers take up dSi across the cell wall, both via diffusion and silicon transporters, to produce biogenic silica (bSi) (hydrated silica – $SiO_2 \cdot nH_2O$), which is used to build internal and external structures (Moriceau et al., 2019; Maldonado et al., 2019). As well as depleting dSi in their local growth environment, the ecological success of silicifiers impacts the cycling of common essential nutrients such as nitrogen, phosphate, and dissolved iron through competition with non silicifiers, and potentially also the cycling



of carbon. Today, the dominant marine silicifiers are diatoms – phytoplankton with a protective opal frustule (silica shell) that mitigates grazing loss (Van Tol et al., 2012) in addition to a relatively fast growth rate (Banse, 1982), enabling them to potentially out-compete other phytoplankton. However, the spatio-temporal distribution of diatoms and their ability to dominate the ecosystem depends on a number of both environmental pressures,

particularly dSi availability, and underlying key metabolic trade-offs, such as control over frustule size, balancing predation vs. buoyancy, and/or optimizing their photosynthetic apparatus in light-intensive areas where excess energy must be dissipated (Hendry et al., 2018; Assmy et al., 2013; Lavaud et al., 2004). The huge number of different possible combinations of trade-offs and marine environments may be behind the evolution of the estimated 30,000 - 100,000 current species worldwide (Mann and Vanormelingen, 2013). As a result of their

potential to dominate ecosystems, the diatom genera is thought to be responsible for approximately 40% of global mean primary production in today's ocean (Field et al., 1998). The different cellular nutrient to carbon ration of diatoms compared to other phytoplankton (O'donnell et al., 2021), together with the potential for the dense protective opal frustules to modify the mean depth at which carbon (and nutrients) can be returned to the water column from sinking biogenic material, implies that a complete representation of the ocean's 'biological pump'

requires that we account for the marine cycle of silica (Wilson et al., 2012).

It is possible to represent rates of nutrient (and carbon) uptake from the ocean surface and subsequent export of solid (and dissolved) biogenic matter in models implicitly as a direct function of the ambient environment such as temperature, light, nutrient availability (Maier-Reimer and Hasselmann, 1987). Such an approach was used in box models (Ridgwell et al., 2002) and more recently in Earth system models of intermediate complexity (EMICs)

that seek to simplify as many elements of the Earth system (especially the atmosphere) as possible for numerical efficiency (M et al., 2002; Weber, 2010). However, the biogenically-induced flux modelling approach is limited both when tasked with exploring events regarding the evolution of ecosystem complexity as ecosystems are not resolved (i.e. plankton diversity is not considered), or in respect to the details of seasonal productivity cycles and species successions and 'blooms', as standing biomass becomes a key state variable that creates temporal lags in

the response of biological export to changes in the ambient physical and chemical environment. In response, model approaches have been developed to resolve this, albeit along a broad spectrum of complexities (Kwiatkowski et al., 2014). At one end, simple 'NPZD' (N - dissolved inorganic nitrogen, P - phytoplankton, Z - zooplankton and D - detritus) models (Kriest et al., 2010) are able to reproduce the variability of the mean ecosystem by simulating the effects of limiting factors (e.g. nutrient limitation), but fail to constrain potentially

important biogeochemical processes and feedbacks associated with the biological pump due to their simplicity (Yool et al., 2013). Beyond this, in terms of complexity, models look to include multiple (plankton) functional types ('PFT's) to better resolve fundamental biogeochemical functions, including those less sensitive to environmental perturbation (Friedrichs et al., 2007; Quere et al., 2005). However, PFTs are generally explicitly based on the observed characteristics of modern plankton, potentially impacting their potential application to past

climates (Ward et al., 2018; Falkowski et al., 2004), because the relationships between species, ecosystems, and environment continually evolve through time, such as the diversification of diatoms in the Cenozoic and their increasing dominance of dSi uptake (Conley et al., 2017). To address this in turn, "trait-based" approaches that focus on the governing rules of diversity as opposed to imposing a specific and restricted diversity, have been devised (Follows and Dutkiewicz, 2011; Follows et al., 2007). Besides requiring fewer total number of parameters

to be specified, trait-based approaches allow a greater resolution of diversity. However, they also require the



identification of the underlying trade-offs that govern species competition and coexistence (Kiørboe et al., 2018). Currently, allometric relationships are generally assumed to regulate these trade-offs, in which physiological and ecological traits can be linked to organism size (e.g. Mullin et al., 1966). Assuming then that these allometric relationships are consistent through time (or at least, rather more conserved than individual species themselves),

trait-based approaches should be comparatively independent of the geological period to which they might be applied.

In the case of the EMIC cGEnIE – a global biogeochemical cycles (ocean circulation and primary climate feedbacks) model designed for addressing paleo questions (Ridgwell et al., 2007), Ward et al. (2018) added a trait-based ecosystem, "EcoGEnIE" which explicitly accounts for the growth of plankton with traits assigned

based on size and function. The paleo utility of now being able to simulate potential ecosystem structures (and associated marine biogeochemical cycles) of the past was demonstrated in Wilson et al. (2018). Here, we build on this earlier work and present an update to the EcoGEnIE 1.0 framework by introducing a diatom phytoplankton functional group (including their allometric relationships) together with a marine silicon cycle – EcoGEnIE 1.1 and tuned the model using Latin hypercube model parameter sampling (Section 3). We then evaluate how the

results of our model ensemble with diatoms compares with global observations and the previous version of EcoGEnIE (Sections 4 and 5). Finally, we discuss how including diatoms into EcoGEnIE provide a critical tool to address scientific questions concerning the biogeochemical and ecological influence of the silicon cycle and the role of diatoms on the carbon cycle on million-year time scales. We start (Section 2) by describing the properties of the cGEnIE Earth system model (e.g., the marine biogeochemical components most relevant to

simulating marine ecology). This section also includes the specific parameterisations employed in the ecosystem component EcoGEnIE including the diatom incorporation.

## 2 The cGEnIE Earth system model

### 2.1 Ocean (-atmosphere) physics

The underlying climate component in the configuration of cGEnIE used here comprises a 3-D frictional geostrophic ocean model coupled with a 2-D energy moisture balance model (EMBM) and a dynamic-thermodynamic sea ice model (Marsh et al., 2011). We employ cGEnIE on a 36×36 longitude vs. latitude grid of

equal area (equal divisions in longitude and the sine of latitude), with ocean depth resolved across 16 vertical layers, that have increasing thickness varying from 80.8 m at the surface to a maximum of 765 m at depth. The continental configuration and ocean bathymetry, together with calibrated parameters controlling ocean, atmosphere, and sea-ice physics, are from Cao et al. (2009).

It should be noted that the continental configuration and tuned physics parameter set that we adopt here is,

different from that of Ward et al. (2018). Because the depth of the mixed layer is critical to calculating mean light penetration and hence photosynthetic rates, Ward et al. (2018) included a mixed layer scheme (Kraus and Turner, 1967). Here, in order to retain the same traceable representation of global ocean circulation as Cao et al. (2009) which formed the basis of the development of a variety of new biogeochemical cycles in cGEnIE (Crichton et al., 2021; Reinhard et al., 2020; Van De Velde et al., 2021), we calculate the mixed layer depth

using the same scheme as employed by Ward et al. (2018) to calculate photosynthetic rates, but do not influence



the ocean circulation. Finally, we also prevent photosynthesis under sea-ice (in practice, in each grid cell, light availability is scaled by the ice-free fraction), which was not adopted in Ward et al. (2018). We quantify and discuss the impact of physics vs. ecosystem structure in contrasting the projections and fit to observations of EcoGEnIE 1.0 vs. EcoGEnIE 1.1 later.

**2.2 Ocean biogeochemical cycling framework**

The BIOGEM module provides the framework for ocean-atmosphere biogeochemical cycling, including regulating air-sea gas exchange as well as the transformation and partitioning of biogeochemical tracers within the ocean. As configured here, BIOGEM accounts for the biogeochemical cycling of carbon, phosphate, oxygen,

carbon (Ridgwell et al., 2007), plus iron (Tagliabue et al., 2016), and a previously devised parameterization of opal dissolution in the water column (summarized below and in the supplemental material) in order to complete the ocean silicon cycle in conjunction with the new ECOGEM diatom addition.

For the iron cycle, we took the preindustrial (year 1850) dust field of Albani et al. (2016) to provide dissolved iron input at the ocean surface, and carried out a brief parameter calibration of the 2 key iron controlling

parameters – the mean global (flux-weighted) iron solubility, and the scaling factor for the scavenging rate of free (non-ligand bound) iron by sinking particulate organic matter in the water column. This was in the form of a 2D parameter ensemble of iron solubility vs. scavenging rate with the resulting simulated 3D distribution of total dissolved iron in the ocean (i.e. free iron plus ligand-bound iron) compared to observations (Tagliabue et al., 2016). We tuned these parameters using the cGEnIE phosphate and iron limitation marine biogeochemical

cycle configuration of Tagliabue et al. (2016) and implemented it in the Cao et al. (2009) configuration of ocean circulation as employed here. We simply then applied these two parameter values to our EcoGEnIE 1.1 configuration. The only differences with respect to the iron cycle parameterization used in Ward et al. (2018) are then: (1) the dust field of Albani et al. (2016) rather than Mahowald et al. (1999), (2) a mean global solubility of dust-delivered iron of 0.244 % as opposed to 0.201 % (partly due to the overall lower dust fluxes of Albani et al.

(2016) vs Mahowald et al. (1999), and (3) a small reduction in the scavenging rate scaling (0.225 vs. 0.344 in Ward et al. (2018).

To complete the ocean silica cycle, opal must dissolve in the water column and at the seafloor, and silica be released back into solution (dSi). The treatment of how sinking biogenic solid silica (bSi) dissolves in the water column follow Ridgwell et al. (2002), which used a simple quasi-empirical scheme that took into account the

degree of ambient opal under saturation and evaluated against sediment trap observations. Note that in this current paper, we do not attempt to calculate the fractional preservation of opal in accumulating sediments at the seafloor, but instead impose a simple benthic 'closure' term and reflect biogenic matter reaching the bottom of the ocean.

     **2.3 Ecological structure**


The ecological extension of the cGEnIE model – 'EcoGEnIE' – consists of highly configurable plankton community (Ward et al., 2018). It defines series of functional groups and respective size classes. Originally, EcoGEnIE 1.0 described and evaluated two types, zooplankton and phytoplankton, which were each delineated



into 8 size classes. A mixotroph functional group was also coded, but not described and evaluated in Ward et al.
(2018) (but was instead employed in a derived model in Gibbs et al. (2020)).

For EcoGEnIE 1.1, we implemented a diatom functional group, coded as a microphytoplankton that has a dSi
nutrient assimilation requirement and associated trade-offs (Tréguer et al., 2018; Follows et al., 2007). In
addition to diatoms, we also differentiated the generic phytoplankton in EcoGEnIE 1.0 into 2 derived
phytoplankton functional types characterized by different photosynthetic rate coefficients: "picoplankton" and
"eukaryotes", based on the trait-based modelling of Dutkiewicz et al. (2020), as shown in Table 1. As per
Dutkiewicz et al. (2020), plankton < 3 μm have an increase in growth rate with size, whereas anything larger
sees a decrease. This is how picoplankton and eukaryotes are differentiated in the model. In total, the EcoGEnIE
1.1 plankton community comprises 4 functional groups (diatoms, picoplankton, eukaryotes and zooplankton).

**Table 1.** Plankton functional groups and sizes in EcoGEnIE 1.1.

| $j$ | Functional type | ESD (μm) |
|---|---|---|
| 1 | Diatom | 2 |
| 2 | Diatom | 20 |
| 3 | Diatom | 200 |
| 4 | Picoplankton | 0.6 |
| 5 | Picoplankton | 2 |
| 6 | Eukaryote | 20 |
| 7 | Eukaryote | 200 |
| 8 | Zooplankton | 6 |
| 9 | Zooplankton | 20 |
| 10 | Zooplankton | 200 |
| 11 | Zooplankton | 2000 |

We configure the assumed size structure of the members of the ecosystem in a more targeted way relative to
Ward et al. (2018). Specifically, we choose to simplify the number of size classes (4 zooplankton instead of 8),
and ensure all non-zooplankton can be grazed upon (this is not the case in Ward et al. (2018), where the largest
phytoplankton is the same size as the largest zooplankton).We tested the implications of assuming the same
0.6μm to 1900μm across 8 size classes range in "EcoGEnIE 1.1y". The assumed functional group and size
structure configuration of ECOGEM focused on in this paper is summarized in Table 1. Overall, these changes
create a more diverse plankton community relative to EcoGEnIE 1.0.

**2.4 Diatom physiology**

The new parameterizations associated with the incorporation of diatoms in ECOGEM are described as follows.
State variables (nutrient resources, plankton biomass and organic matter) in EcoGEnIE 1.1 follow the same
equations in EcoGEnIE 1.0, as described in the supplemental material.

**2.4.1 Size-dependent traits**

Power-law functions of organismal volume ($V = \pi[\text{Equivalent spherical diameter}]^3/6$) define a given size-
dependent parameter ($p$). $V_0$ is a reference value of 1 μm³. Values $a$ and $b$ are size scaling coefficients.

$$p = a(\frac{V}{V_0})^b \qquad (1)$$



In contrast to EcoGEnIE 1.0, which applies a unimodal photosynthetic uptake rate relationship for all phytoplankton, each phytoplankton functional group within the EcoGEnIE 1.1 population possesses specific rates as per (Dutkiewicz et al., 2020), as shown in Table 2.

### 2.4.2 Diatom extension

As per the other plankton functional groups in the model, diatom biomass ($B_{Diat}$) varies over time as a balance between a growth term that depends on the uptake rate (V), and limitations by light, temperature and nutrients plus loss terms (grazing and mortality), which are fully described in the supplemental material.

$$\frac{dB_{Diat}}{dt} = V_{Diat} \cdot B_{Diat} - ((Grazing_{Diat} \cdot Palatability_{Diat}) + Mortality_{Diat}) \qquad (2)$$

We used commonly defined trait and trade-offs of diatoms to characterise their competitiveness relative to other plankton, ensuring they would mimic the superiority that is seen in observations (Tréguer et al., 2021). Diatom defined traits include higher growth rate $P_C^{max}$ (See growth curve; (Dutkiewicz et al., 2020)), dSi limitation through associated nutrient parameters, and reduced palatability, which is defined by a unitless parameter that modifies the relative grazing palatability on a group (Table 2). This reduced palatability relative to the rest of the community accounts for diatoms' competitive ability to mitigate grazing losses via their protective frustules (Zhang et al., 2017). Diatoms produce both organic matter and bSi, which get exported into the water column through diatom mortality or detritus from feeding, where zooplankton only incorporate the organic part.

## 3 Model tuning

We tuned both the new diatom-specific model parameters as well as a selection of other ECOGEM parameters central to how phytoplankton behaviour in general is controlled. These are listed in Table 2. We compared model results with global ecological and diatom observations (see section 3.2).

### 3.1 Tuning method

The parameters, whose values we explored in the tuning process, include minimum and maximum nutrient quotas, maximum uptakes rates, and nutrient affinities. We tested a range of values derived from the literature as summarized in Table 2. We also tuned diatom palatability to best simulate diatom's grazing protection. We kept (Ward et al., 2018)'s parameter values for phosphate maximum uptake rate and the cellular carbon quotas as preliminary sensitivity experiments showed little sensitivity on biogeochemical output (mean oxygen concentration, export production, etc.) when exploring values around the previously well-constrained estimated values (e.g. studies seen in Table 2). We then used Latin hypercube sampling (Mckay et al., 2000) to generate a 550-member ensemble sampling uniformly across the 18 model parameters we had identified as critical to controlling ecosystem dynamics (and hence of marine biogeochemical cycles). For each ensemble member experiment, we calculated a M-score (Watterson, 2015) to gauge model-data fitness with greater values

**Table 2.** List of ECOGEM parameters selected for tuning and the range of tested values and cited literature (Ward et al., 2018; Ragueneau et al., 2006; Dutkiewicz et al., 2020; Edwards et al., 2012).



| Parameter | Symbol | Tested range | Best run | Units | References |
|---|---|---|---|---|---|
| **Quota** | $Q_P^{min}$ | $10^{-3} - 10^{-2}$ | $2.7 \times 10^{-3}$ | mmol P (mmol C)$^{-1}$ | |
| | $Q_P^{max} : Q_P^{min}$ | 1 - 10 | 8.0 | mmol P (mmol C)$^{-1}$ | Ward et al. (2018) |
| | $Q_{Fe}^{min}$ | $5 \times 10^{-7} - 1.5 \times 10^{-6}$ | $0.7 \times 10^{-6}$ | mmol Fe (mmol C)$^{-1}$ | |
| | $Q_{Fe}^{max} : Q_{Fe}^{min}$ | 1 - 10 | 6.0 | mmol Fe (mmol C)$^{-1}$ | Ward et al. (2018) |
| | $Q_{Si}^{min}$ | $0.01 - 0.1$ | 0.04 | mmol Si (mmol C)$^{-1}$ | |
| | $Q_{Si}^{max} : Q_{Si}^{min}$ | 1 - 10 | 9.4 | mmol Si (mmol C)$^{-1}$ | Ragueneau et al. (2006) |
| **Max photosynthetic rate** | $P_{C_{diatom}}^{max}$ | $3.9 V^{-0.08}$ | - | mmol C (mmol C)$^{-1}$ d$^{-1}$ | Dutkiewicz et al. (2020) |
| | $P_{C_{other}}^{max}$ | $2.2 V^{-0.08}$ | - | mmol C (mmol C)$^{-1}$ d$^{-1}$ | Dutkiewicz et al. (2020) |
| | $P_{C_{pico}}^{max}$ | $0.9 V^{0.08}$ | - | mmol C (mmol C)$^{-1}$ d$^{-1}$ | Dutkiewicz et al. (2020) |
| **Max uptake rate** | $V_{Fea}^{max}$ | $5 \times 10^{-5} - 2 \times 10^{-4}$ | $1.7 \times 10^{-4}$ | mmol Fe (mmol C)$^{-1}$ d$^{-1}$ | |
| | $V_{Feb}^{max}$ | -0.5 - -0.25 | -0.13 | | Ward et al. (2018) |
| | $V_{Sia}^{max}$ | $0.01 - 0.1$ | 0.07 | mmol Si (mmol C)$^{-1}$ d$^{-1}$ | |
| | $V_{Sib}^{max}$ | $0.01 - 0.1$ | 0.03 | | Ragueneau et al. (2006) |
| **Nutrient affinities** | $\alpha_{Pa}$ | $0.5 - 1.5$ | 0.94 | m$^3$ (mmol C)$^{-1}$ d$^{-1}$ | |
| | $\alpha_{Pb}$ | -0.5 - -0.25 | -0.44 | | Ward et al. (2018) |
| | $\alpha_{Fea}$ | $0.15 - 0.2$ | 0.18 | m$^3$ (mmol C)$^{-1}$ d$^{-1}$ | |
| | $\alpha_{Feb}$ | -0.5 - -0.25 | -0.26 | | Ward et al. (2018) |
| | $\alpha_{Sia}$ | 1 - 5 | 4.8 | m$^3$ (mmol C)$^{-1}$ d$^{-1}$ | |
| | $\alpha_{Sib}$ | -0.5 - -0.25 | -0.40 | | Edwards et al. (2012) |
| **Grazing protection** | $\phi_{diatom}$ | $0.3 - 0.8$ | 0.93 | | |




representing better performance:

$$M = \frac{2}{\pi}\arcsin\left[\frac{\sum_{i=1}^{n}\frac{(M_i - O_i)^2}{n}}{\sigma_m^2 + \sigma_o^2 + (\mu_m - \mu_o)^2}\right] \tag{10}$$

Here, the model (m) and observational (o) value in the $i^{th}$ ocean grid points (cell) out of a total n grid points are represented by $M_i$ and $O_i$ respectively, with mean square error described in the numerator. Mean and variance

are denoted $\sigma^2$ and $\mu$. M-score therefore is non-dimensional and is value between 0 and 1, with higher values indicating better model-data performance.

**3.2 Observations**

We assessed how successful EcoGEnIE 1.1 was in generating realistic oceanic biogeochemistry by comparing

model outputs to observations from the World Ocean Atlas 2013 (WOA13) climatological datasets of dissolved oxygen, phosphate and dSi (Garcia et al., 2013). These were in the form of annual averages and were re-gridded onto the model grid prior to statistical comparison. We also visually contrasted modelled chlorophyll concentrations to an average from 1997 to 2002, measured by the SeaWiFs satellite (Seawifs).

**3.3 Model experiments**


We created an initial 20,000-year long spin-up of the complete system (iron and silica cycles plus diatom-enabled ECOGEM plankton ecology) with the default values from EcoGEnIE 1.0 for the 18 key plankton related parameters. Each of the 550 ensemble members were then run for 2,000 years continued from the same spin-up experiment, with the (annual average) output derived from the final year of the experiment. Tests of

longer integration times showed that little further change occurred in biogeochemical indicators (oxygen, phosphate <1% change etc.) beyond 2,000 years.

**4 Results**

**4.1 Model ensemble and parameter set choice justification**


We first look at the performance of the 550 model members towards observations. Figure 1 shows the results from the tuning ensemble, ordered by average M-score for dissolved oxygen, phosphate, and silica results. Figures 2 and 3 have the same format but zoom into just the best 50 performing M-scores. It is clear that there are apparent trade-offs in the overall M-score statistics, with for example high $O_2$ M-scores coinciding with

lower $PO_4^{3-}$ M-scores and vice versa. A scatter plot included in the supplemental material further highlights this. We also observe a similar relationship between the mean ocean oxygen concentration (observed concentrations of ~162 µmol l$^{-1}$) and export production (which involves oxygen consumption during remineralisation). Thus, while one might select the overall best M-score experiment when identifying a tuned parameter set to go forward with, other considerations might be important to take into account.






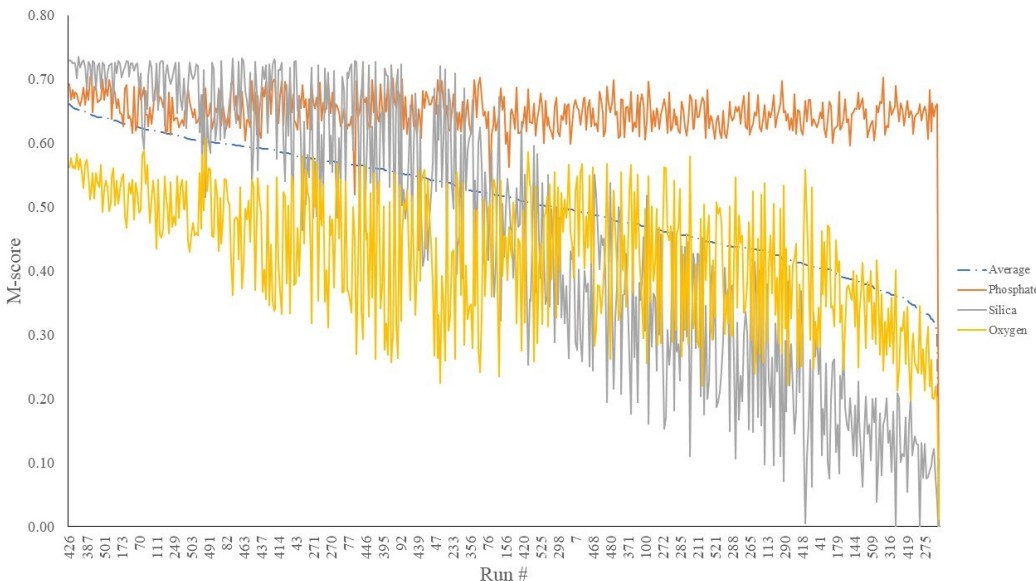

**Figure 1.** M-scores of O₂, SiO₂ and PO₄ of 550-run ensemble.

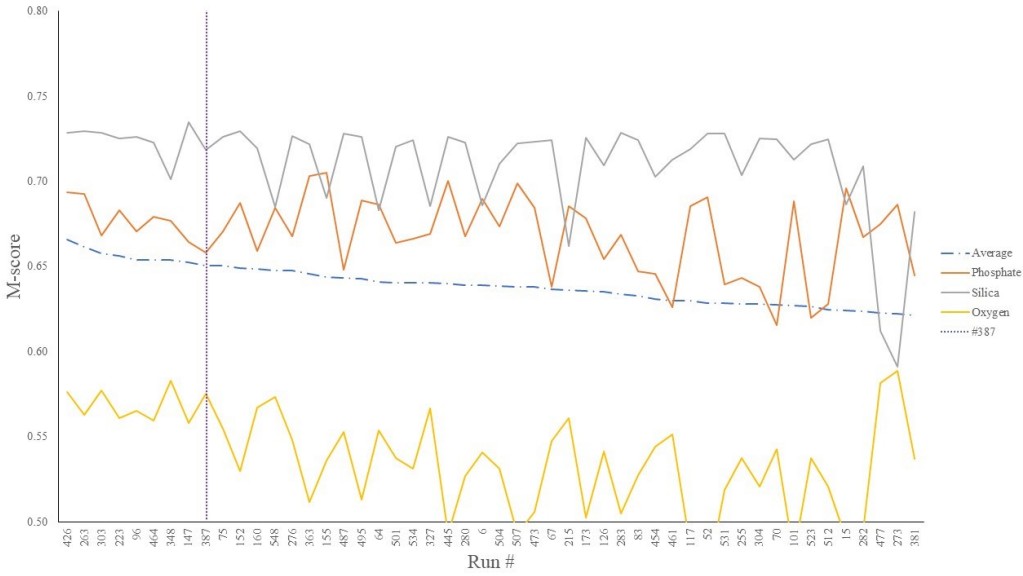

**Figure 2**. Top 50 M-score runs from the 550-run ensemble.



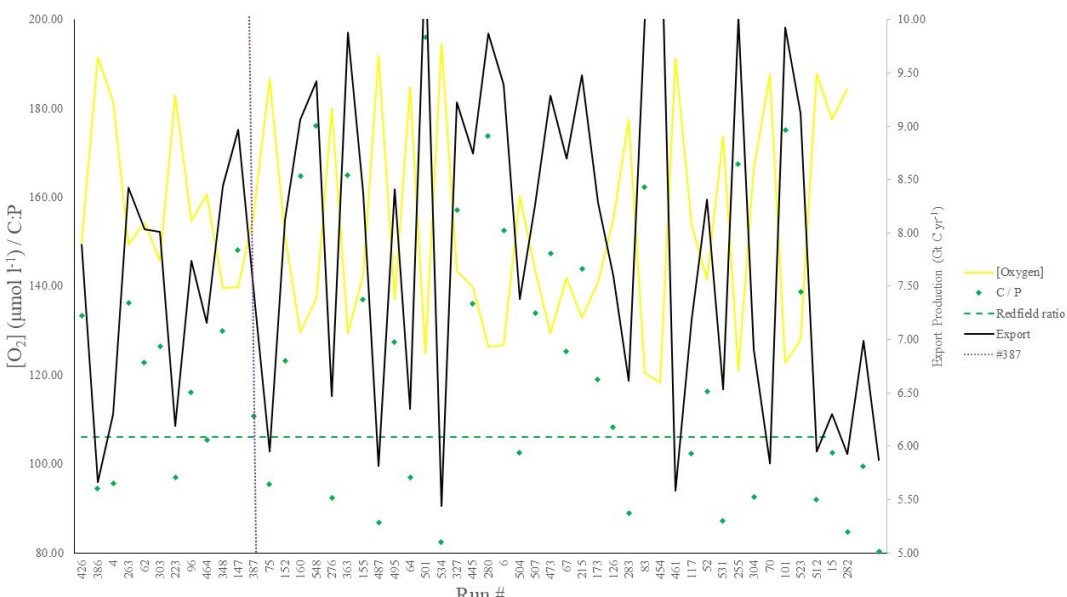

**Figure 3.** POC export (mol m-2 yr-1), C:P export ratios and mean oxygen concentration (μmol l-1) of the top 50 runs.

We chose our "best" run (run #387) through balancing M-score trade-offs and selecting realistic oxygen concentrations (promoted by reasonably low export). Because phosphate M-scores are high across the ensemble, we prioritised runs with a high $O_2$ M-score despite the aforementioned trade-off. We also picked the run with a carbon-to-phosphate export ratio close to the Redfield ratio of 106 and recent inverse models' estimations of ~ 105 - 113 (Wang et al., 2019; Matsumoto et al., 2020; Teng et al., 2014). This characteristic favoured #387

(global C:P of 111) over the best silica performing run (run #96), which had a C:P of 116. While this study primarily concerns diatoms and the silicon cycle, we are also tuning the ecosystem as a whole, thus a realistic global C:P of export is imperative. Whilst run #464 has a similarly good C:P and slightly higher average M-score than run #387, our pick has opal export value (103 Tmol Si yr-1) within estimates (Table 3) of 100 – 140 Tmol Si yr-1 (Nelson et al., 1995) .

Run #387 also produces a global total POC export of 7.5 Pg C yr-1 (Fig. 3, Table 3), falling well within estimates of 4 -12 Pg C yr-1 (Devries and Weber, 2017; Henson et al., 2011; Dunne et al., 2005). Mean oxygen concentration produced by this iteration is also acceptable at 156 μmol kg-1, close to the 162 estimates, whilst other high statistically scoring runs produced values beyond this range (e.g. run #426). Such attributes give run #387 an average M-score of 0.65, within the top 15 performing runs (Fig. 2 and Fig. 3).

**4.2 Biogeochemical variables**



Overall, EcoGEnIE 1.1 captures the zonal contrast in phosphate concentrations between low and high latitudes (> 2 µmol P l$^{-3}$ towards the poles with ~ 0 equatorially, Fig. 4). The model underestimates phosphate (WOA13 records ~1 µmol P l$^{-3}$) in equatorial and margin upwelling environments, a known issue for cGEnIE due to its
simplified physics and low spatial resolution. The model-data comparison is also not strictly like-for-like, because in re-gridding higher vertical resolution WOA to the model grid, elevated subsurface concentrations become averaged into the re-gridded 'surface' layer. So, some of the apparent model under-estimation of surface values could be due to the data re-gridding. However, the model estimates dSi concentrations in the equatorial upwelling reasonably (~ 20 µmol Si l$^{-3}$), although it is below the elevated concentrations in the surface northern
Pacific (~ 40 vs >50 µmol Si l$^{-3}$, Fig. 5).

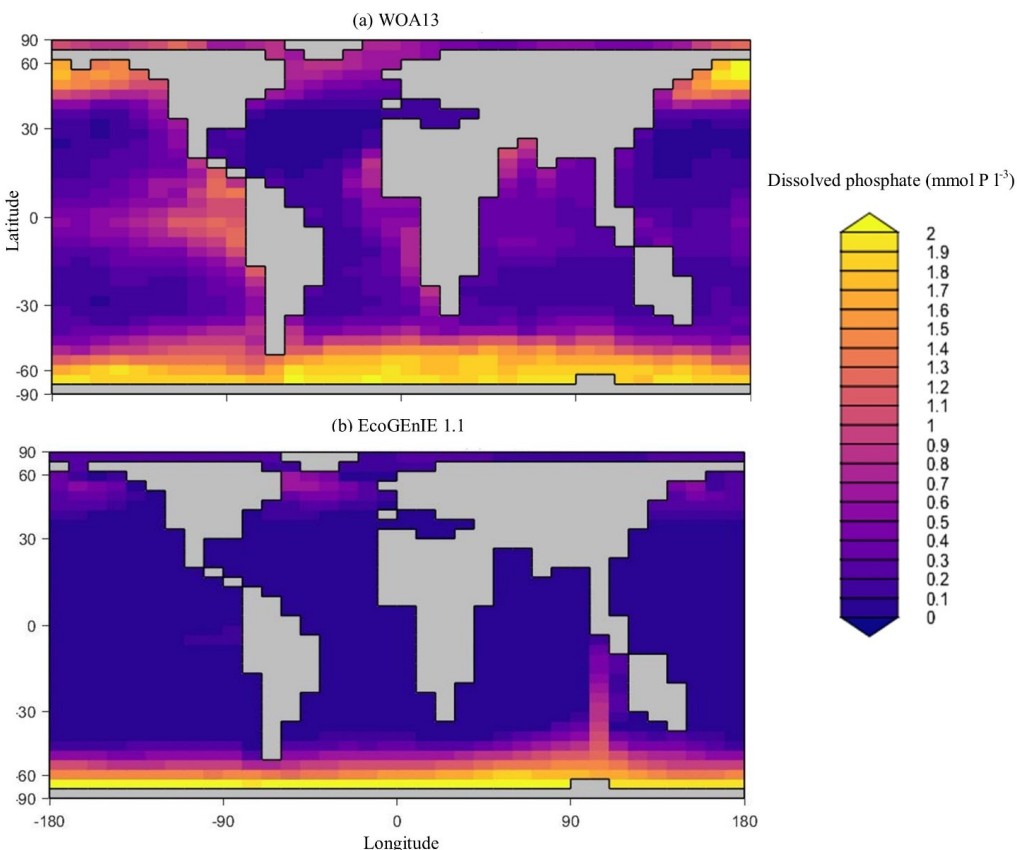

**Figure 4.** Surface concentrations of dissolved phosphate (mmol P l$^{-3}$).



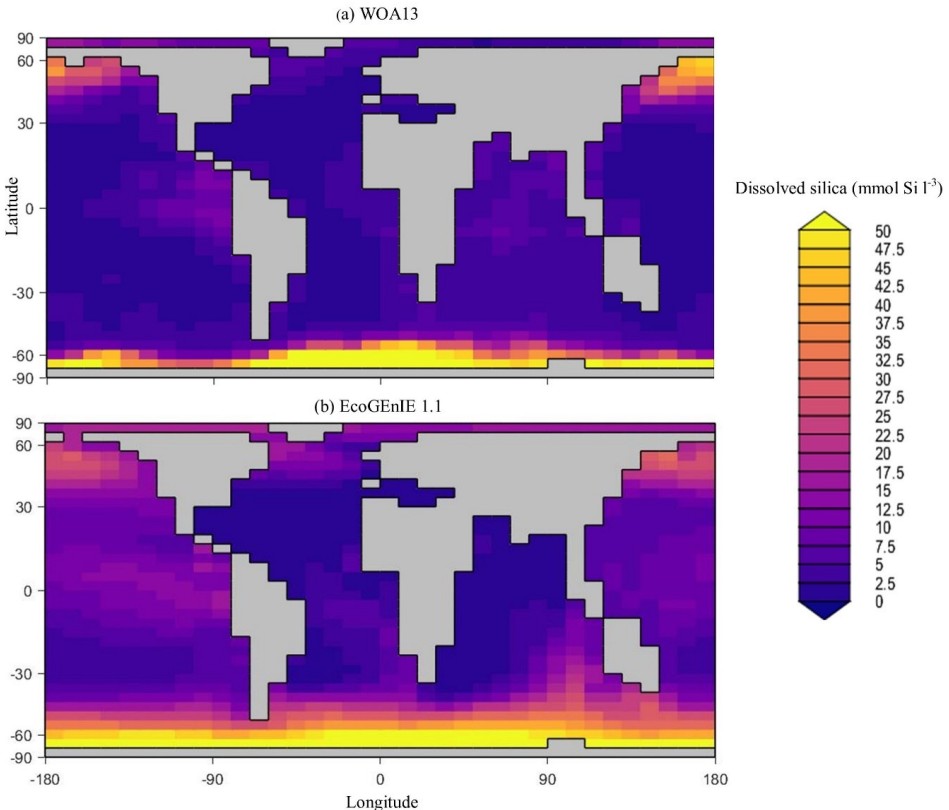

**Figure 5.** Surface concentrations of dSi **(a)** and **(b)**, mmol $SiO_2$ $l^{-3}$).

Through the ocean, EcoGEnIE 1.1 reasonably captures the main features of the vertical biogeochemical tracer distributions in each of the three main ocean basins (Atlantic, Indian, and Pacific). Like in the observations, the model captures the transport of dissolved oxygen (~300 µmol $O_2$ $l^{-1}$) at depth through the North Atlantic and Southern Ocean via deep-water masses (Fig. 6). The model performs as well for phosphate concentrations (Fig. 7), with a slight underestimation in the intermediate northern Atlantic, tending towards 0.5 µmol P $l^{-1}$ vs 1 µmol P $l^{-1}$ at 1000 – 3000m depth. These complement diminished export and remineralisation in EcoGEnIE 1.1, allowing greater uptake of dissolved nutrients at intermediate depths. The highest concentrations of phosphate (3 mmol P $l^{-3}$) in the equatorial Indian ocean (2000 – 4000 m) seen in the data are limited to >2 km depths in the model, likely due to restricted resolutions at depths and the smaller size of the Indian basin. The same trends (discrepancies to WOA13 are most notable at the greatest depths) are observed in the model for dSi (Fig. 8), which is generally represented accurately across the three model ocean basins approaching 0 µmol Si $l^{-1}$ in the surface and peaking at approximately 120 µmol Si $l^{-1}$ at depths (below 1000m in the Pacific and Indian ocean).



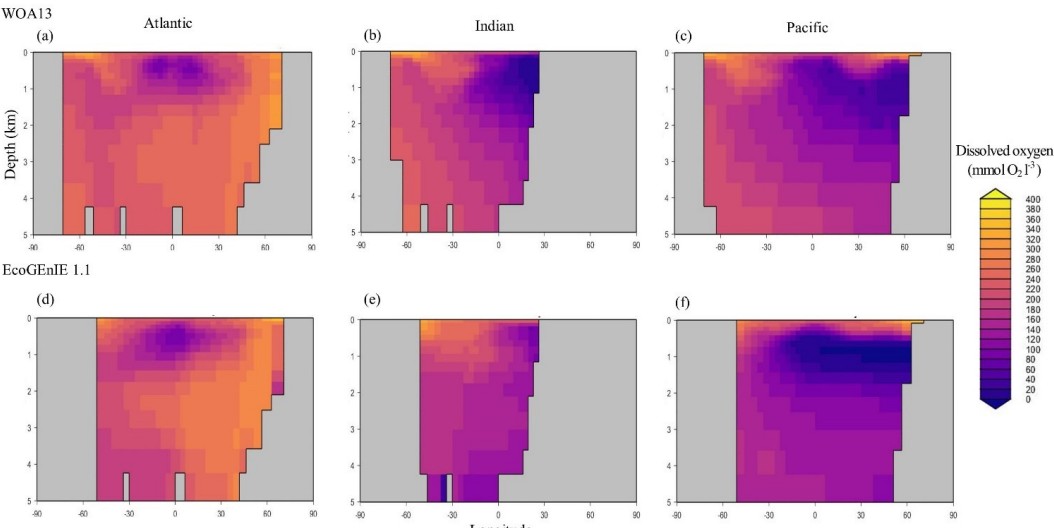

**Figure 6.** Zonally averaged vertical distribution of dissolved oxygen for the best EcoGEnIE 1.1 (**(d)-(f)**) run ($\mu$mol
$O_2\ l^{-1}$) compared to WOA13 (**(a)-(c)**).

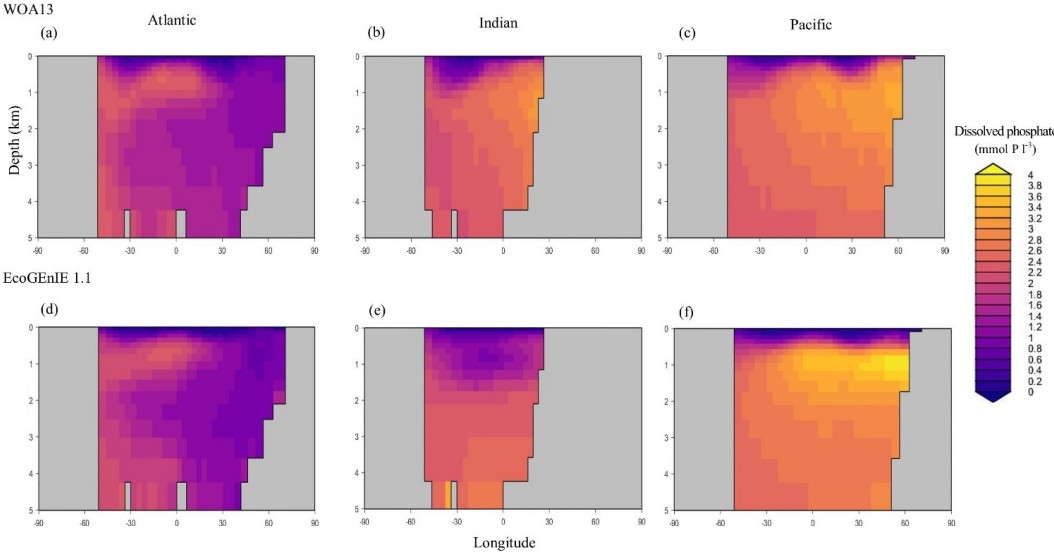

**Figure 7.** Zonally averaged vertical distribution of dissolved phosphate ($\mu$mol P $l^{-1}$) of EcoGEnIE 1.1 (**(d)-(f)**)
compared to WOA13 (**(a)-(c)**).



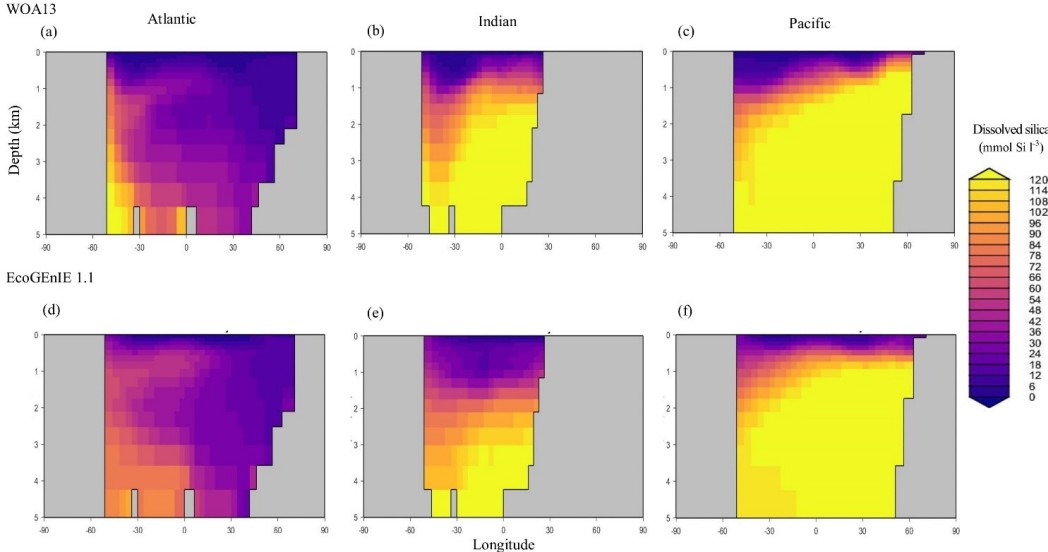


**Figure 8.** Zonally averaged vertical distribution of dSi (μmol Si l⁻¹) of EcoGEnIE 1.1 (**(d)-(f)**) compared to WOA13 (**(a)-(c)**).

### 4.3 Ecological variables


We also assessed performance of the tuned model relative to observations in chlorophyll from the SeaWiFs satellite (Seawifs), as well as export production relevant metrics such as the organic matter C:P ('Redfield') ratio. We additionally assess carbon biomass distributions in our configured plankton community.

Firstly, EcoGEnIE 1.1 chlorophyll biomass compares generally well with satellite estimates, peaking at ~1 mg
Chl l⁻³ in the high latitudes and equatorially. EcoGEnIE 1.1 tends to have more widespread peak Chl values than in the satellite images, with lower Chl in the subtropics and prominent Chl in the Southern Ocean (Fig. 9). However, it is known that satellite observations can underestimate concentrations in the high latitudes (Dierssen, 2010). This could help explain some of the model disagreement in the Southern Ocean. For the Arctic, the sign of the model-data mismatch is reversed and is more likely to be primarily due to the limited model resolution in
this basin, reflecting restricted circulation in the model and/or poor seasonal sea-ice cover.

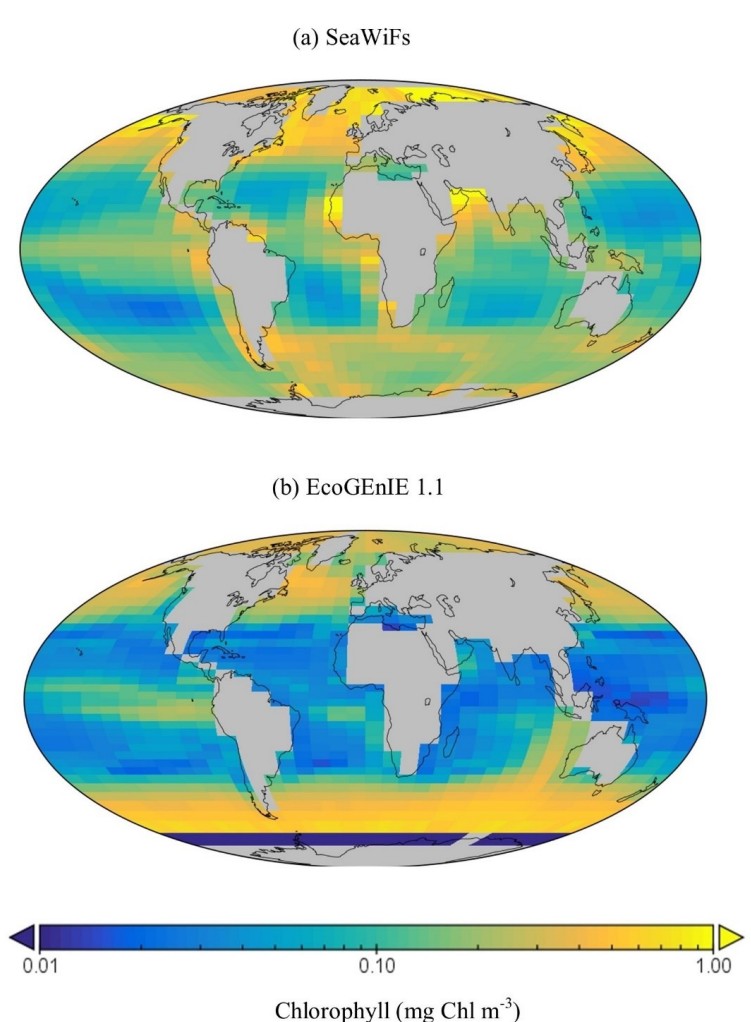

**Figure 9.** Satellite-derived **(a)** and modelled **(b)** surface chlorophyll a concentration (mg Chl m$^{-3}$).


**Table 3.** Performance of the three renditions of EcoGEnIE (EcoGEnIE, NoDiatom and EcoGEnIE 1.1) and their correlation coefficients to WOA13 data. Two additional runs are shown, one in which our new physics are applied to EcoGEnIE 1.0 (EcoGEnIE 1.1x) and another where the EcoGEnIE 1.0 plankton population replaces that 345    configured in EcoGEnIE 1.1 (EcoGEnIE 1.1y).



| | EcoGEnIE 1.0 | EcoGEnIE 1.1x | EcoGEnIE 1.1y | NoDiatom | EcoGEnIE 1.1 | Estimates |
|---|---|---|---|---|---|---|
| **Ecological configuration / Plankton** | **1.0 / 1.0** | **1.0 / 1.0** | **2.0 / 1.0** | **2.0 / 2.0 (minus diatoms)** | **2.0 / 2.0** | |
| **Ocean biogeochemical configuration** | **1.0** | **2.0** | **2.0** | **2.0** | **2.0** | |
| **$O_2$ M-score** | 0.51 | 0.40 | 0.46 | 0.57 | 0.57 | |
| **$PO_4$ M-score** | 0.62 | 0.64 | 0.63 | 0.66 | 0.66 | |
| **$SiO_2$ M-score** | - | - | - | - | 0.72 | |
| **Average M-score** | 0.56 | 0.52 | 0.55 | 0.62 | 0.65 | |
| **$[O_2]$ / $\mu mol\ kg^{-1}$** | 140 | 104 | 128 | 174 | 156 | ~ 162 |
| **POC export / $Pg\ C\ yr^{-1}$** | 11.3 | 10.6 | 9.0 | 6.6 | 7.5 | 4 – 12 |
| **Opal export flux / $Tmol\ Si\ yr^{-1}$** | - | - | | - | 103 | 100 – 140 |
| **Export C:P** | 138 | 150 | 124 | 100 | 111 | 106 |

Global annual mean POC export in the model is 7.5 Pg C $yr^{-1}$ (Table 3), which is within the estimated range of 4 - 12 Pg C $yr^{-1}$ (Devries and Weber, 2017; Henson et al., 2011; Dunne et al., 2005). Spatially, the modelled POC and oapl export reaches relatively high values (>4 mol C $m^{-2}$ $yr^{-1}$ and >1 mol Si $m^{-2}$ $yr^{-1}$ respectively) in the

subpolar regions such as the Southern Ocean, the North and East equatorial Pacific, and exhibits relatively low values (<1 mol $m^{-2}$ $yr^{-1}$) in the subtropical gyres and high polar latitudes (Fig. 10a). The latter is due to sea-ice formation in the Southern Ocean, which in the model is assumed to prevent light penetration and hence limits production, while the Arctic low production is due to limited model resolution preventing the model to capture adequately the basin dynamics as mentioned above.

The global C:P export ratio is approximately 111 in our preferred model calibration (Table 3), with higher ratios (> 120) in the subtropical gyres and low ratios (~ 90) in the subpolar and upwelling regions (Fig. 10b). This distribution, at least visually, agrees with previous estimates (Teng et al., 2014; Tanioka et al., 2022) with the exception of the North Atlantic, which has previously been observed to have extremely high values (~200). One reason that the model struggles to produce these high C:P ratios because it does not include a nitrogen cycle

(and nitrogen fixation), thus regions where nitrogen may be at low or high concentrations (e.g., North Atlantic) may possess unrealistic C:P ratios.



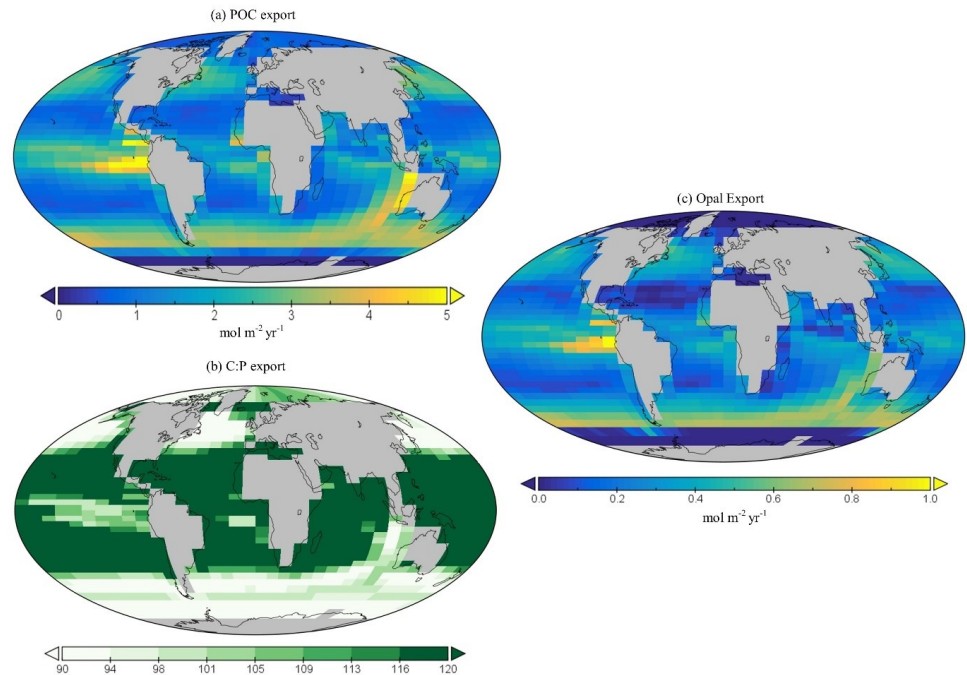

**Figure 10.** Global POC **(a)** and opal export (**(c)**, mol m⁻² yr⁻¹). Global surface distribution of carbon to phosphorus ratio for export of particulate organic matter **(b)**.

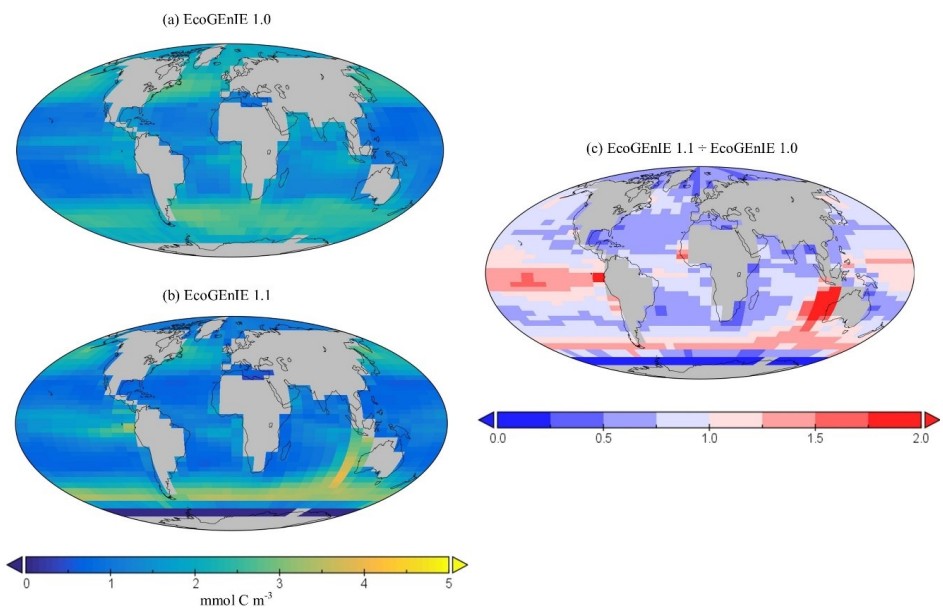




**Figure 11.** Surface concentrations of total carbon biomass of EcoGEnIE 1.0 **(a)** and EcoGEnIE 1.1 (**(b)**, mmol C m$^{-3}$). Panel **(c)** depicts the relative increase or decrease of EcoGEnIE 1.1 from EcoGEnIE 1.0 for vertical fluxes of particulate organic carbon (mmol C m$^{-2}$ d$^{-1}$).

Total model carbon biomass of all plankton resembles chlorophyll distribution with high values in the nutrient-rich regions and low in the subtropical gyres (Fig 11b). This biomass is the sum of the 4 plankton groups with picoplankton omnipresent whilst eukaryotes inhabit more distinct niches with higher biomass (Fig. S1). Zooplankton biomass mimics the ones of their preys, as they are configured to prey on plankton at least 10 times smaller than themselves (Fig. S2).

The spatial distribution of diatoms (all size classes combined) in EcoGEnIE 1.1 agrees with previous estimates (Tréguer et al., 2018), with high concentration in the productive regions (e.g. equatorial upwellings, subpolar regions) and peakings in the Southern Ocean at ~ 1 mmol C m$^{-3}$ (Fig. 12d). Diatoms contribute to 17.9% of total carbon biomass in the model and 7% of exported carbon. Of the 3 different size classes parameterized in the model (Table 1), the smallest (2 μm) is the most omnipresent and is abundant across all dSi-enriched regions –

the Southern Ocean, equatorial upwelling zones, and North subpolar region (Fig. 12a). Their larger counterparts (20 μm) dominate in the subpolar and equatorial upwelling regions (Fig. 12b) and boasts a greater peak biomass – approximately double that of the 2 μm size class (0.37 versus 0.21 mmol C m$^{-3}$). The 200 μm diatom size class is further restricted in geographical extent, consistent with as diatoms increase in size in the model, they become increasingly restricted to dSi-enriched regions, most notably to the Southern Ocean (Fig. 12c). The relative

carbon biomass distribution of the 2 and 20 μm diatoms carbon biomass is depicted in Figure 13. The Southern Ocean presents a dominance of diatoms within the larger size class, with over twice the carbon biomass than the 2 μm class. In contrast, Equatorial upwelling regions are characterized by a somewhat equal size distribution between 2 μm and 20 μm, with the 2 μm class having slightly greater eminence. All diatom size classes are virtually absent within the subtropical gyres and low nutrient regions.



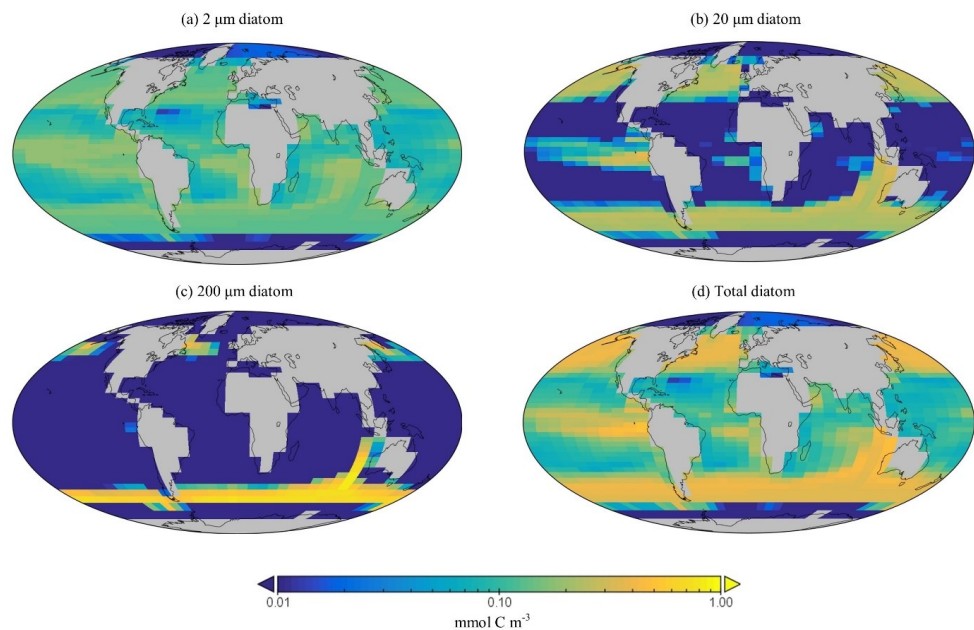

**Figure 12.** Surface concentrations of carbon biomass for each diatom size class (**(a)** – **(c)** mmol C m⁻³) and their summed biomass **(d)**.

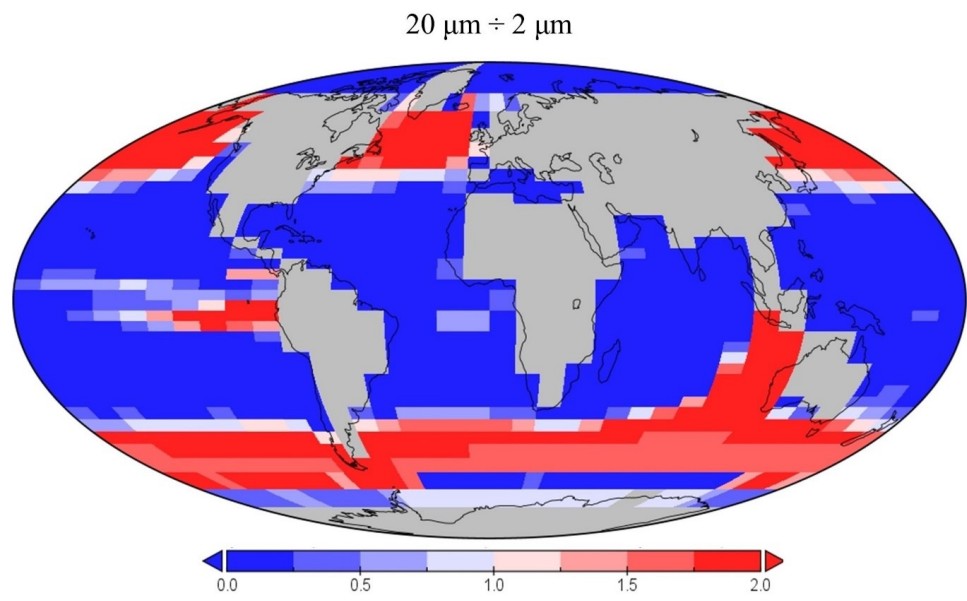

**Figure 13.** The relative eminence of diatoms in the 20 µm size class compared to the 2 µm class.



**5 Discussion and conclusions**

In this section, we assess and discuss how the projections of EcoGEnIE 1.1 compare to EcoGEnIE 1.0, what has changed and why, paying particular attention to the impact of changing the configuration of the underlying

ocean circulation component as well as adding a diatom functional type. We then discuss the capability of EcoGEnIE 1.1 in simulating diatom biogeography within size classes, and then finish with a paleo application perspective.

**5.1 EcoGEnIE 1.0 vs EcoGEnIE 1.1**

We contrast EcoGEnIE 1.1 model outputs with the previous and original ecological version EcoGEnIE 1.0 (Ward et al., 2018), the latter including 8 size classes of phytoplankton and zooplankton and with no diatom functional type as well as employing a different iron cycle and ocean physics. In order to help separate out these factors, we also contrasted EcoGEnIE 1.1 with a NoDiatom run, configured in an identical fashion to EcoGEnIE 1.1 but with no diatoms (same ocean circulation and iron cycle) and two other iterations (EcoGEnIE 1.5x and

1.5y, described later).

Relative to EcoGEnIE 1.0, EcoGEnIE 1.1 performs better for all regarded biogeochemical tracers with higher M-score (oxygen, phosphate, silica, Table 3). EcoGEnIE 1.1 mean oxygen concentration is much more realistic than in EcoGEnIE (156 versus 140 μmol O2 l$^{-1}$), which was due to higher export production rates (7.5 versus 11.3 Pg C yr-1), causing higher respiration. Basin profiles in EcoGEnIE 1.0 (Fig. 14) exhibit unrealistic elevated

and widespread dysoxia in the low latitude and northern regions of the Indian Ocean. Such an issue was likely due to the enhanced export, leading to greater oxygen consumption at intermediate depths.

Despite a significantly lower global export in EcoGEnIE 1.1, specific regions have higher export relative to EcoGEnIE 1.0 (Fig. 15a). Productive regions, notably equatorial upwellings, exhibit elevated export in EcoGEnIE 1.1. Similar patterns are also seen in NoDiatom export distribution which has lower production

equatorially than EcoGEnIE 1.1 (Fig. 15b). This suggests that regionally, the greatest impact has come from changing the ocean biogeochemical configuration. That said, the absence of diatoms in NoDiatom intuitively results in a Southern Ocean with less export production than EcoGEnIE 1.1.

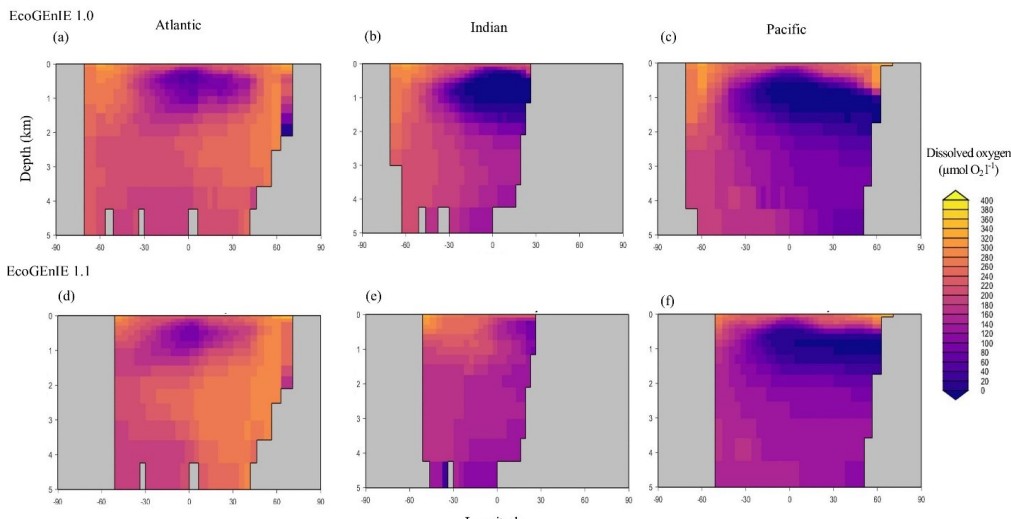



**Figure 14.** Zonally averaged vertical distribution of dissolved oxygen for the best EcoGEnIE 1.1 **((d)-(f))** run ($\mu$mol $O_2$ $l^{-1}$) compared to EcoGEnIE 1.0 **((a)-(c))**.

(a) EcoGEnIE 1.1 ÷ EcoGEnIE 1.0

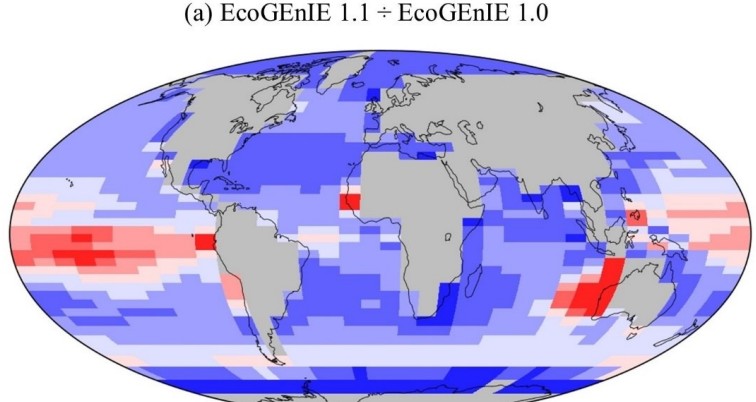

(b) NoDiatom ÷ EcoGEnIE 1.0

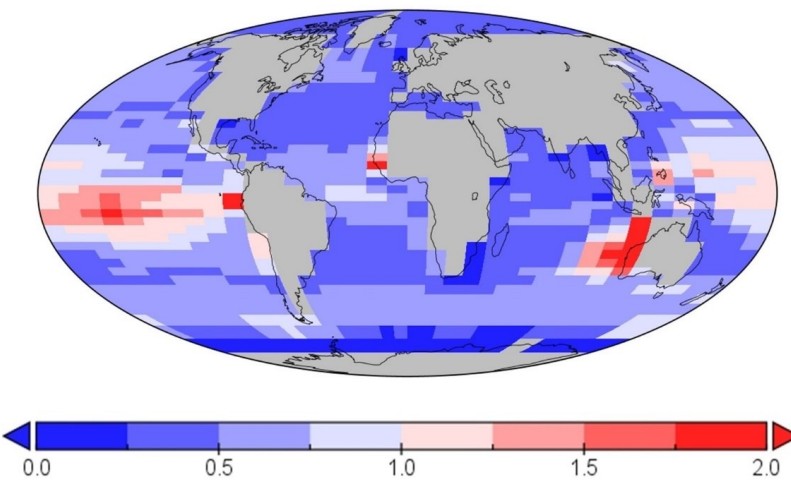


**Figure 15.** The relative increase or decrease of EcoGEnIE 1.1 from EcoGEnIE **(a)**, and NoDiatom from EcoGEnIE 1.0 **(b)**, for vertical fluxes of particulate organic carbon (mmol C $m^{-2}$ $d^{-1}$).

Another improvement apparent in EcoGEnIE 1.1 and NoDiatom is the C:P export ratio, which is pushed

significantly closer to the Redfield value of 106 likely thanks to the tuning of Qmin and Qmax values performed in this study to produce more realistic stoichiometries. Such results imply that the steps taken in to refine the model have produced more realistic and comprehensive biogeochemical interactions within the water column.




We suspect that the retuning of C:P export ratio to ~111 helped EcoGEnIE 1.1 to produce more favourable basin profiles.

There is a clear improvement from EcoGEnIE 1.0 in the distinction between low biomass in subtropical gyres and high in higher latitudes (Fig. 11), which can be attributed to the modified ocean physics in NoDiatom and EcoGEnIE 1.1. Equatorial chlorophyll biomass has a noticeable increase from the original rendition, producing results far closer to satellite estimates, likely a result of a thriving diatom population. The introduction of diatoms moving from NoDiatom and EcoGEnIE 1.1, is most notably felt in the Southern Ocean, likely due to

the high concentrations of dSi which allows the greatest diatom productivity. Overall, there is marginal change in the output of EcoGEnIE 1.1 and NoDiatom runs (relative to the change of EcoGEnIE 1.0 to 1.1), suggesting that in the absence of diatoms, other phytoplankton take advantage of vacant niches that diatoms would otherwise occupy. With our trait-based approach enabling size diversity amongst functional types, it is intuitive that plankton of the same sizes to the diatom classes would make up the difference with regards to the primary

production deficit (i.e., size is the master trait).

The differences between EcoGEnIE 1.0 and 2.0 arise both due to the developments in (adding phytoplankton functional groups, changing size structure) and tuning of ECOGEM, the switch in underlying ocean biogeochemical configuration, and the re-tuning of the primary physiological parameters in ECOGEM. Table 3 includes runs where the Ward et al. (2018) ecosystem was combined with our new physics (EcoGEnIE 1.5a) and

another where the EcoGEnIE 1.0 size class population replaces that seen in Table 2 (EcoGEnIE 1.5b). We found that EcoGEnIE 1.1x only achieved slightly improved model correspondence to observations for phosphate, with the oxygen M-score drastically decreasing. There is, however, a decrease in export production (11.3 vs 10.6 Pg C yr$^{-1}$), suggesting the change in ocean circulation configuration helped improve this result. EcoGEnIE 1.1y also shows slight improvements to phosphate M-score, it is likely that the ecosystem tuning (which is notably tuned

to EcoGEnIE 1.1's plankton community) somewhat contemplates the EcoGEnIE 1.0 plankton community due to the similar size range. Once we introduce our functional groups (NoDiatom) coupled with the new oceanic circulation, it is evident how much the results improve (M-scores, export etc.). Adding the diatom functional group (and thus ecologically enabling the silica cycle) then improved the M-scores further with reasonable opal export.

**5.2 Relative diatom distribution**

EcoGEnIE 1.1 produced diatom populations in which the smallest size class (2 μm) are the most dominant, a result that agrees with the relatively few observational estimates that are available, as we will discuss next.

Genomic ribotype reads and *in situ* plankton recording in the northern Atlantic observe smaller diatoms to be
more abundant than larger ones (Barton et al., 2013; Malviya et al., 2016), a characteristic we also find in EcoGEnIE 1.1, although relative north Atlantic abundance within the model (2 μm diatoms relative to the 20 μm diatoms) is two orders of magnitude greater than these recordings. However, studies such as these tend to be poorly constrained via instrumentation, as meshes cannot sample plankton < 20 μm, potentially resulting in a significant part of the plankton community remaining unrecorded, thus EcoGEnIE 1.1 produces seemingly
unrealistic relative abundances. Sensitivity testing in the model suggests that the proportion of these size classes depends on the uptake rate for dSi, $V_{sia}^{max}$ (Fig. 16). As $V_{sia}^{max}$ increases, the ratio of carbon biomass attributed to



20 µm compared to 2 µm biomass decreases. This is intuitive – the allometric relationships within functional groups result in larger plankton becoming less competitive as their nutrient quotas and uptake rates increase metabolic demand. There is a notable absence of the 200 µm diatom class in the northern Atlantic, despite recordings by Barton et al. (2013), suggesting EcoGEnIE still struggles to constrain larger plankton.


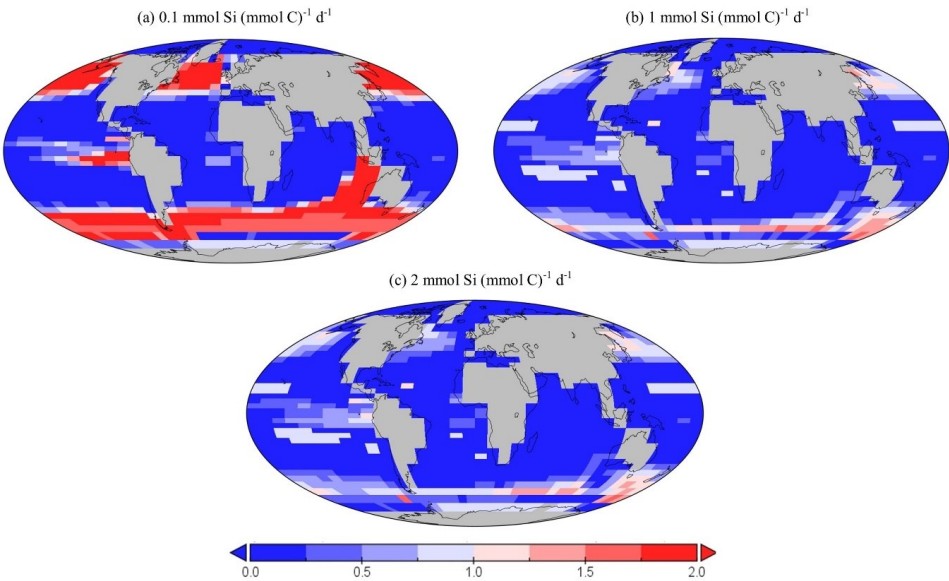

**Figure 16.** Sensitivity testing for dSi uptake rates of diatoms. Panels **(a)** to **(c)** depict the relative eminence of diatoms in the 20 µm size class compared to the 2 µm class. Values above 1 therefore indicate a region dominated by larger diatoms.


The difficulties associated with assessing the ecological performance of EcoGEnIE 1.0 persist in EcoGEnIE 1.1. With plankton biomass restricted to the upper layer (80.8 m) of the GEnIE ocean circulation model grid, direct comparisons to data collected in situ can be somewhat difficult, with satellite observations inconsistent in their definition of the depth of the surface layer. *In situ* measurements of diatom distribution tend to opt for ribotype

reads of size classes along expedition transects (Malviya et al., 2016); we are consequently restricted to inferences of regional patterns amongst classes as opposed to direct comparisons of global population. Although there has been a successful ecological extension of the model, there are persistent limitations that ultimately arise from the efforts to trade off model complexity and user convenience.

**5.3 Conclusion**


This paper builds on the EcoGEnIE 1.0 model of Ward et al. (2018), which developed a size-based formulation of plankton ecology and embedded this in an Earth system model of intermediate complexity. We expanded the model to include a diatom and other phytoplankton functional groups and hence enable the marine silica cycle to be simulated. We not only tuned the model parameters for diatoms, but also re-tuned the most critical



physiological parameters in the ecosystem model framework, identifying a parameter configuration that performed best towards observations of biogeochemical tracers and ecological variables.

The EcoGEnIE 1.1 model successfully incorporated diatoms as a functional type, enabling as a limiting resource. The competitive nature and success of diatoms in captured in the model as a prevalent group in our configured community (~ 20% of total biomass). With this new extension, there is a potential for further study

regarding the ecological success of diatoms during future and past climatological perturbation and their role in the biological pump. For example, with these additions, this model could be utilised to explore the Cenozoic evolution of diatoms and their ongoing influence over the silicon cycle, long-term silica cycling (e.g. residence times) and their associated proxies (Conley et al., 2017; Tréguer et al., 2021). This study also acts as an example of the adaptability of the EcoGEnIE model, encouraging those looking to incorporate additional functional

groups into the framework.

**6 Code availability**

The code for the version of the 'muffin' release of the cGEnIE Earth system model used in this paper, is provided at https://www.seao2.info/cgenie/docs/muffin.pdf. Configuration files for the specific experiments presented in

the paper can be found via the DOI: 10.5281/zenodo.7643548 (newest version). Details of the experiments, plus the command line needed to run each one, are given in the readme.txt file in that directory. All other configuration files and boundary conditions are provided as part of the code release. A manual detailing code installation, basic model configuration, tutorials covering various aspects of model configuration, experimental design, and output, plus the processing of results, can be found at https://www.seao2.info/cgenie/docs/muffin.pdf.

**Author contributions**

ANB tuned the model under the supervision of FM with respect to the ecosystem. The iron biogeochemistry was developed by AR and JDW. KH and DC provided their expertise regarding the silicon cycle. BW provided insights regarding the development of EcoGEnIE. All authors contributed to the writing of this paper.

**Acknowledgments**

We thank Rhiannon Jones for early discussions regarding the silicon cycle modelling. This research has received funding from the European Research Council (ERC) under the Horizon 2020 Research and Innovation Programme (#833454 DEVOCEAN). KRH thanks European Research Council ERC-ICY-LAB (#678371).

FMM thanks NERC for its funding (NE/N011708/1, NE/V01823X/1, NE/X001261/1). JDW acknowledges support from the AXA Research Fund.



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
