# Peer review of "A diatom extension to the cGEnIE Earth system model – EcoGEnIE 1.1"

_EGUsphere, 2022_

## Author Comment (AC2)

We thank the reviewer for their constructive comments on the manuscript. We have addressed all points in a revised manuscript and in responses below. Reviewer comments are shown in bold text and our responses in standard text, with actioned responses in green.

The major changes are as follows:

- More details and clarity have been implemented regarding the diatom trade-off presentation. This included figures within the supplementary material and further descriptive text in the manuscript.
- Further explanation of the EcoGEnIE sensitivity experiments (i.e., 1.1x and 1.1y, now renamed 1.1_phys and 1.1_phys_eco respectively), with added figures to provide context into the implications of the model development, namely ocean physics and ecosystem modification. This includes new comparisons of biogeochemical/ecological tracers produced by the original and new renditions of EcoGEnIE.

The minor points have also been addressed, including typos and suggested elaboration of certain arguments.

We believe these additions have improved the manuscript's clarity and hope the editor is in agreement.

**Referee**/Response/Actioned Response

**Major point: The paper revolves around the implementation of diatoms in the Genie Earth System Model. As such, the paper will be the future reference for the diatom sub-model. However, some aspects of the description are incomplete (see specific comments). A particular missing point is the grazing protection/palatability. Where does "phi_diatom" enter the model in equation (7) in the supplementary? And why is the "best run" value in table 2 (0.93) outside the tested range (0.3 – 0.8)? Finally, it would be nice to see a better presentation of the "trade-offs", i.e., the differences between the diatoms and the pico/other phytoplankton, e.g., by graphs of the key parameters with cell size.**

We thank the reviewer for suggesting the need for more details in the model description, which will benefit future users of this rendition of EcoGEnIE. Many of the topics in this major point comment summarise the minor point comments, which are addressed individually further down this document.

We have now removed the phi_diatom notation (using instead "*Palatability*") to avoid confusion it with the prey-switching term. Palatability as mentioned in equation (2) in the manuscript is multiplied by the term described in equation (7) of the supplementary. It is a unitless parameter describing the relative palatability (opposite of grazing protection) of a diatom relative to other prey (where palatability for other plankton = 1 in this study).

We have corrected the tested range presented in the table to 0.3 – 1; 0.8 was a typo.

Finally, we have added the figure (S6, RC1) below into the SI together with explanatory text to further detail the trade-offs associated with diatoms. The figure outlines the main diatom trade-offs defined in the model, which include reduced palatability and higher maximum photosynthetic rate. The final trade-off is the obligate requirement diatoms have for Si, also noted in Table S1.

**Figure RC1** Graphs of key differences ((a) maximum growth rate, (b) palatability) between diatoms and other plankton within EcoGEnIE 1.1.

[Figure]

**Minor points:**

**Section "Grazing" in supplementary. "The 'prey-switching' term is optional …".
Please specify whether this term is applied in the present implementation?**

We have clarified this in the supplementary material. Like in the EcoGEnIE 1.0 version
(Ward et al., 2018), EcoGEnIE 1.1 accounts for active switching by setting pre-switching
term(s) to 2.

**Table S1: "Diatom trade-offs". It is a little unclear where the trade-offs refer to:
higher Pmax than what? I suppose it is higher than the phytoplankton group.
Another benefit is the higher maximum photosynthetic rate (should also be
mentioned line 200).**

We have added clarification in the table and the main text. $P_{max}$ is higher for diatoms
relative to other phytoplankton groups of similar size as shown in Table S1 and Figure
RC1 that present the equation for the maximum photosynthetic rate (Pmax) for the
modelled phytoplankton groups. We have updated the text that defines this benefit now
referring to "photosynthetic growth rate" to be fully transparent.

**Supplementary "temperature limitation". What are the values of A and Tref? Are
they the same as for the other phytoplankton group?**

This is correct. We have added the values of A and Tref in the text, with A = 0.05 $^oC^{-1}$
and Tref = 20$^oC$. The model temperature limitation of metabolic processes is consistent
across every functional group (same values of A).

**Line 162: "and associated trade-offs". Since the inclusion of diatoms is novel, it
would be nice to referring to the trade-offs and dSi assimilation requirements
already here.**

We have replaced "associated trade-offs" with "higher Pmax, lower palatability and
obligate requirement for dSi" in the manuscript. We have also referenced Table S1 and
added the sketch above to add more description on diatom's trade-offs.

**Line 174: "We tested the implications of assuming the same 0.6μm to 1900μm
across 8 size classes range in "EcoGEnIE 1.1y". This sentence is unclear; what
was the purpose of that test?**

We have added further explanation of these tests in the manuscript. The purpose of
EcoGEnIE 1.1x and 1.1y (**now renamed 1.1_phys and 1.1_phys_eco for clarity**) is to
assess the individual impacts of updating the physics and adding ecosystem functional
groups from EcoGEnIE 1.0, which had a different size structure than EcoGEnIE 1.1

Thus, where a rendition of EcoGEnIE = Plankton population + Ecosystem configuration
+ Physics configuration, we can define these tests as (with selected rendition as
subscript):

EcoGEnIE 1.1_phys = Plankton $_{EcoGEnIE\ 1.0}$ + Ecosystem$_{EcoGEnIE\ 1.0}$ + Physics$_{EcoGEnIE1.1}$

EcoGEnIE 1.1_phys_eco = Plankton $_{EcoGEnIE\ 1.0}$ + Ecosystem$_{EcoGEnIE\ 1.1}$ + Physics$_{EcoGEnIE1.1}$

Thus 1.1_phys corresponds to EcoGEnIE 1.1 but with 1.0's plankton and ecosystem configuration, and 1.1_phys_eco is EcoGEnIE 1.1 with just 1.0's plankton. This allows comparisons of size-diversity range (1.0: 0.6 - 1900μm vs 1.1: 2 – 2000 μm) and functional diversity (1.0: 2 functional groups vs 1.1: 4 functional groups).

**Line 188: "In contrast to EcoGEnIE 1.0, which applies a unimodal photosynthetic uptake rate relationship" How does this affect the model? It would have been nice if the implication of the different photosynthetic rates had been discussed in the interpretation of the results.**

This refers to the minor point above. We tested for this effect by comparing EcoGEnIE 1.1_phys with 1.1_phys_eco which show the effect of moving from a unimodal scheme to individual photosynthetic rates with the same physics configuration. Adding functional groups helps the model to perform better towards its oxygen cycling and export production, with C:P export closer to the Redfield ratio (C:P of 124 vs 150).

**Line 194. It creates confusion to use the symbol "V" for both volume and max uptake rate.**

We thank the reviewer for bringing that to our attention, as maximum uptake rates within the model code are $V_{max}$. We will keep uptake rate to "$V$" and change volume to "$Vol$" so that consistency is kept with the manuscript and model code. This difference will be noted in the manuscript.

**Line 203: Please explain "bSi"**

bSi is already defined in the introduction (Line 37). It is defined as the silica utilised by silicifiers to build internal and external structures. We have changed this to opal here for the context of export production.

**Lines 308-310: "The highest concentrations of phosphate (3 mmol P l-3) in the equatorial Indian ocean (2000 – 4000 m) seen in the data are limited to >2 km depths in the 310 model, likely due to restricted resolutions at depths and the smaller size of the Indian basin.". It seems as if the vertical resolution is sufficiently fine to resolve more shallow high concentrations of P. Is it due to too low vertical mixing?**

This is the case, Figure 7e evidently shows there is not enough upwelling in the north Indian ocean to replicate the World Ocean Atlas phosphate. This upwelling issue can be attributed to the mentioned simple ocean circulation physics and low resolution of the model (especially for the smaller Indian basin).

**Eq (10). The equation number should be (3).**

We have changed this to the correct equation number.

**Table 2: Does the subscript "other" refer to the "eukaryotes" in table 1?**

This is correct, the Pmax value is described in Table 1 within the supplemental material. We have also removed this row from Table 2 in the manuscript as Pmax is not being tuned.

**Fig.9 : It would have been nice to see Chl-a predictions with "EcoGEnIE 1.0" to assess how the inclusion of diatoms modifies this aspect of the model (if at all).**

We have included Figure RC2 in the revised manuscript. Discussion of the differences of Chl-A in the renditions is outlined in the other minor points regarding Chl-A below.

[Figure]

**Figure RC2** Total chlorophyll biomass for EcoGEnIE 1.0 (mg Chl m$^{-3}$)

**Figure 9b. I was surprised to see the relatively low Chl values in equatorial and eastern boundary upwelling systems. This does not fit well with the high concentrations of diatoms in these regions in figure 12d. It would be relevant to comment on this in the paragraph lines 329-.**There is noticeable low Chl-A in the equatorial upwelling regions, an issue also visible in EcoGEnIE 1.0 (Fig. RC2). However, overall, Figure 9 and 12d (Fig. RC3 and RC4) show similar distributions of Chl-A and total diatom biogeography, with EcoGEnIE 1.1 presenting improved and distinct subtropical gyres from the original rendition. This issue with equatorial upwellings is persistent with EcoGEnIE and is due to the aforementioned simple physics and low resolution of the model – this point has been included in the relevant line.

**Figure RC3** Total chlorophyll biomass for EcoGEnIE 1.1 (mg Chl m⁻³)

[Figure]

**Figure RC4** Total Diatom biomass for EcoGEnIE 1.1mmol C m$^{-3}$

**Line 349. "opal"**

We have corrected this typo.

**Line 370: "Total model carbon biomass of all plankton resembles chlorophyll distribution with high values in the nutrient rich regions and low in the subtropical gyres (Fig 11b)." What about Southern Ocean? Chl-a is high at zones of the Southern Ocean (Fig.9b) where total carbon biomass is low. Why? Fig12c shows higher total diatom biomass in the Southern Ocean, but the total carbon biomass is lower here.**

We thank the reviewer for spotting this, we have corrected this in the revised manuscript with Figure RC4. Figure 12d (total carbon biomass for diatoms) was incorrectly scaled (it should be 0.01 to 1 mmol C m$^{-3}$ like for 12a – 12c, this was not the case). It is now evident that total carbon biomass is higher than the Figure 12c biomass. Regarding total carbon biomass, the scale for Figure 11b (0-5 mmol C m$^{-3}$) is different to the Chl-A plot (0-1 mg Chl m$^{-3}$) which is why it may appear Chl-A is high where total carbon biomass is low.

**Line 417: "Despite a significantly lower global export in EcoGEnIE 1.1, specific regions have higher export relative to EcoGEnIE 1.0 (Fig. 15a)." Do you speculate somewhere the reason for this?**

This change in POC flux distribution is primarily due to the physics update (as shown with the comparison between 1.1x (now 1.1_phys) and 1.0 shown in Figure RC5). The distribution change when switching to our new ecosystem is minimal vs the switch of physics. Figure RC5 shows a POC export difference plot of EcoGEnIE 1.1x (New physics) and EcoGEnIE 1.0. It bears similarity to Figure 14a, suggesting that the physics switch together with the addition of the diatom functional group are primarily responsible for the differences between EcoGEnIE 1.0 and 1.1 (notably the equatorial Pacific). We have included this plot in the supplemental.

[Figure]

1.1_phys / 1.0

**RC5** Difference plot of POC flux distribution between EcoGEnIE 1.1$_{phys}$ and EcoGEnIE 1.0.

**Lines 421-422: "That said, the absence of diatoms in NoDiatom intuitively results in a Southern Ocean with less export production than EcoGEnIE 1.1." How does this compare to Fig11c at the areas of Southern Ocean where we see decrease in vertical POC fluxes in the diatom version?**

A comment has been added at line 497 to clarify this: The southernmost region of the Southern Ocean decreases in EcoGEnIE 1.1 in POC export we observe in Figure 11c can be attributed to the new sea ice module in which growth is no longer enabled at these high latitudes (see previous response regarding sea ice). The decreases above the red band seen in Figure 11c are due to the new physics where subtropical gyres are better defined such that plankton growth is more restricted at these regions/latitudes.

**Lines 437-438: "Equatorial chlorophyll biomass has a noticeable increase from the original rendition, producing results far closer to satellite estimates, likely a result of a thriving diatom population." Do you show anywhere chlorophyll biomass from the original rendition? If yes, add reference here. Otherwise, it would be useful to include it.**

Chlorophyll of the original rendition (Fig. RC3) has been added to the revised manuscript. We have also added a difference plot in the supplemental (Fig. RC6) to demonstrate the regions (Southern Ocean, equatorial upwelling zones) where chlorophyll is elevated from the model development.

Chl-A EcoGEnIE 1.1 / 1.0

[Figure]

**RC6** Difference plot of chlorophyll biomass between EcoGEnIE 1.1 and EcoGEnIE 1.0.

**Lines 487-488: "there are persistent limitations that ultimately arise from the efforts to trade off model complexity and user convenience". This is quite a generic remark; some concrete arguments are missing here.**

We thank the reviewer for their feedback, this sentence has been clarified in the manuscript: Many of the limitations this model's performance can be attributed to the low resolution of the grid system, a feature however that enables the user to perform relatively fast simulations. This can be seen for example in the lower depths of the basin profiles (e.g. Fig. 8) and the high latitudes (e.g. Fig. 10). The plankton recording data discussed in 5.2 is a good example of how the gridded system of the surface ocean restricts concrete comparison of model output and regional data. The limited number of tracers of ECOGEM (due to model simplicity) combined with low-resolution spatial attributes make it difficult to analyse model performance despite how accessible the model can be to run.

**Fig.11b: Why does carbon biomass decrease in the southernmost cells of the Southern Ocean in the diatom version? Is it due to the sea ice module?**

This is correct, the supplementary material outlines the several differences between EcoGEnIE 1.0 and 1.1 (Table S1), which includes a sea-ice light limitation change. Now there is no light under sea ice, phytoplankton cannot grow via photosynthesis, which was not the case for EcoGEnIE 1.0 (see Section 2.1).

**Figure 13 caption. "Eminence"? Do you mean "presence"?**

Done.

References

Dutkiewicz, S., Cermeno, P., Jahn, O., Follows, M. J., Hickman, A. E., Taniguchi, D. A. A., and Ward, B. A.: Dimensions of marine phytoplankton diversity, Biogeosciences, 17, 609-634, 10.5194/bg-17-609-2020, 2020.

Ward, B. A., Wilson, J. D., Death, R. M., Monteiro, F. M., Yool, A., and Ridgwell, A.: EcoGEnIE 1.0: plankton ecology in the cGEnIE Earth system model, Geosci. Model Dev., 11, 4241-4267, https://doi.org/10.5194/gmd-11-4241-2018, 2018.

Wilson, J. D., Monteiro, F. M., Schmidt, D. N., Ward, B. A., and Ridgwell, A.: Linking Marine Plankton Ecosystems and Climate: A New Modeling Approach to the Warm Early Eocene Climate, Paleoceanography and Paleoclimatology, 33, 1439-1452, 10.1029/2018pa003374, 2018.

---

## Author Comment (AC3)

We thank the reviewer for their constructive comments on the manuscript. We have addressed all their points in a revised manuscript and in responses below. Reviewer comments are shown in bold text and our responses in standard text, with actioned responses in green.

The major changes mostly involve providing further justification for methodology choices (i.e. use of datasets), clarification of language choice and expanded explanation for discussed results.

We have also addressed minor points attributed to typos and referencing errors.

We believe these additions have improved the manuscript's clarity and hope the editor is in agreement.

**RC2**: Anonymous Referee #2, 19 Sep 2023

**Referee**/Response/Actioned Response

**This manuscript presents a useful addition to the literature, and should in my view be published after addressing the relatively minor issues identified below.**

**Line 142: "The only differences with respect to the iron cycle parameterization used in Ward et al. (2018) are then: (1) the dust field of Albani et al. (2016) rather than Mahowald et al. (1999), (2) a mean global solubility of dust-delivered iron of 0.244 % as opposed to 0.201 % (partly due to the overall lower dust fluxes of Albani et al. (2016) vs Mahowald et al. (1999), and (3) a small reduction in the scavenging rate scaling (0.225 vs. 0.344 in Ward et al. (2018)." This is helpful information, but requires more justification. Presumably these numbers resulted from model turning, in which case this should be stated, and the objectives of that tuning described.**

The reviewer is correct, these numbers arise from model tuning. The objective of this was to implement a more up-to-date dust field into the model, as the original rendition was tuned to data from 1999. The numbers presented are a result of this change in dataset, this has now been clarified in the manuscript.

**Line 151: "we do not attempt to calculate the fractional preservation of opal in accumulating sediments at the seafloor, but instead impose a simple benthic 'closure' term and reflect biogenic matter reaching the bottom of the ocean." What do you mean by 'reflect' here? Please describe this more completely, what is the fate of this silica?**

We thank the reviewer for pointing out this potentially unclear definition. The silica reaching the seafloor is entirely dissolved, thus the global ocean silica inventory in the ocean remains unchanged and is thus "reflected" from the seafloor instead of being buried. We have changed this word choice in the manuscript for clarification.

**Line 235: What are you using old WOA data? Would updating this make a difference to your tuning, for example with increased data at high latitudes?**

Yes we are using WOA data from 2013, this enables direct comparison to the original EcoGEnIE rendition (now added in the updated manuscript). When testing with more recent WOA datasets we found minimal difference in model performance.

**Line 245: "little further change occurred in biogeochemical indicators (oxygen, phosphate <1% change etc.)" Be more specific - max surface ocean concentrations, global ocean inventory….**

We thank the reviewer, this has been clarified to M-scores i.e. those plotted in Figure 1.

**Line 281: "Mean oxygen concentration produced by this iteration is also acceptable at 156 µmol kg-1, close to the 162 estimates" - please provide a citation for this estimate.**

The average for mean oxygen concentration has been taken from the used WOA dataset. We now mention this in the manuscript.

**Line 286 I don't think that "Overall, EcoGEnIE 1.1 captures the zonal contrast in phosphate concentrations between low and high latitudes" is a very good description. The contrast is in fact between the polar and sub polar regions, excluding the Arctic, and the the rest of the oceans. Low latitude implies latitudes around the equator, where the model performance is poor.**

The reviewer is correct, it has now been clarified here that the contrast is between the polar and the subpolar, and that the contrast at low latitudes (i.e. equator) is due to poor model performance.

**"The model-data comparison is also not strictly like-for-like, because in re-gridding higher vertical resolution WOA to the model grid, elevated subsurface concentrations become averaged into the re-gridded 'surface' layer.". Does the physics of GENIE allow for a meaningful mixed layer to form? If not, there will not be the barrier to nutrients being entrained into the ~80m top level and the comparison seems reasonable? Further, can this argument apply to phosphate, but not to silica, where the results are pretty good, and one needs to instead justify why WOA is lower than your simulation in much of the ocean? Expanding your argument would help people like me who are not very familiar with the consequences of GENIE's highly simplified physics.**

The gridded WOA observation data include observations from the top 80.8m of the ocean, so will include observations from below the true mixed layer when it is shallower (thus shallow regions such as the equatorial Pacific appear elevated in phosphate in the observations), thus the issue is likely a source of bias. This has been addressed in the manuscript.

The mixed layer is only represented in the model in terms of how the light available to the phytoplankton is calculated (see section 3.2.6 of the EcoGEnIE 1.0 model description paper). EcoGEnIE then works out what the chlorophyll concentration would be if it were mixed evenly across this depth, and then works out what the average light level should be across that depth with that level of chlorophyll. This scheme is not used for nutrients or biomass, thus the model does not have a "meaningful" mixed layer in that sense.

Both phosphate and silica have very good M-scores but indeed silica is visibly better. This can be attributed to the mechanisms in which they are regulated in the model, notably during deep ocean cycling. Phosphate is regulated by decomposition of sinking material (traded off with $O_2$, see response to trade-off comment below), whereas silica is regulated by the dissolution of sinking opal. As decomposition requires respiration ($O_2$ consumption), improving model performance of phosphate (e.g. intermediate Pacific) would likely push oxygen to less realistic concentrations. Silica is not influenced by this trade-off thus higher M-scores are achievable.

**"The spatial distribution of diatoms (all size classes combined) in EcoGEnIE 1.1 agrees with previous estimates (Tréguer et al., 2018)…" This is a key criteria for this manuscript, and needs to be expanded on. Currently I believe you only present results from your model here (please make figure captions more descriptive so this is not ambiguous) - this needs to be contrasted with other estimates or datasets in a more robust way, acknowledging the challenges around data availability. At present the verification of the distribution of the key new PFT**

**you have added to the model is "… agrees with previous estimates (Tréguer et al., 2018), with high concentration in the productive regions (e.g. equatorial upwellings, subpolar regions) and peakings in the Southern Ocean at ~ 1 mmol C m" within the results section, and a comment on the relative size distribution in the discussion. Where observation based comparison is simply not possible because of limitations to available observations, explain this to the reader.**

The reviewer is correct that observation-based comparison is simply not possible because of limitations to available observations, which is also referred to in the 5.2 section when discussing plankton recording techniques. We do indeed only present diatom biogeography from the modelling and have made this clearer in the figure captions (e.g. Figure 16's caption has been changed to clarify these results are all model-borne). We have also added further explanation with the paragraph discussing spatial distribution of diatoms that alludes to the difficulties associated with ecological datasets (e.g. Maredat) and that the more robust verification methods are through biogeochemical tracers i.e M-scoring.

**Line 377: "peakings" should be "peaking"**

Changed.

**What hypotheses are stimulated by the trade-offs seen in your M-scores? This behaviour is telling you something about the system or the limitation to the modelling approach - can you propose any suggestions?**

We hypothesise that the M-score trade-off between oxygen and phosphate is due to the limitation of the model. Phosphate concentrations are regulated by decomposition of sinking matter, which influences the extent of respiration and thus the amount of oxygen consumed. As previously mentioned, achieving more accurate phosphate concentrations would lead to elevated oxygen levels and would require modification of the $O_2$:P stoichimetry. It is also likely to be an issue due to insufficient ocean mixing with, $O_2$ not being replaced to counteract diminishing concentrations. We have added a sentence in section 4.1 and the supplemental to suggest the reason for this trade-off. It is worth noting that achieving accurate oxygen performance is a common issue amongst global biogeochemical models.

**General:**

**- ensure consistent formatting of references.**

We have used the EndNote output style file for referencing and ensured they are consistent with GMD's submission requirements.

**Minor specific points:**

**Line 51 "ration" to "ratio"**

Done.

**Line 598 sort out the reference and its citation in the text.**

The citation has been removed as this sentence was removed in a re-edit.

**Table 1. Define ESD in the caption.**

This is equivalent spherical diameter and has now been defined in the caption.